green chemistry/materials science/chemical engineering

waste, adsorption, bio-based, molasses, congo red, remediation

**Author for correspondence:**
Nontipa Supanchaiyamat
e-mail: nontsu@kku.ac.th

This article has been edited by the Royal Society of Chemistry, including the commissioning, peer review process and editorial aspects up to the point of acceptance.

# Graphitic mesoporous carbon-silica composites from low-value sugarcane by-products for the removal of toxic dyes from wastewaters

Intuorn Janekarn[1], Andrew J. Hunt[1], Yuvarat Ngernyen[2], Sujittra Youngme[1] and Nontipa Supanchaiyamat[1]

[1]Materials Chemistry Research Center, Department of Chemistry and Center of Excellence for Innovation in Chemistry, Faculty of Science, Khon Kaen University, Khon Kaen, 40002, Thailand
[2]Department of Chemical Engineering, Faculty of Engineering, Khon Kaen University, Khon Kaen, 40002, Thailand

AJH, 0000-0003-3983-8313; NS, 0000-0002-5031-4281

Highly porous carbon-silica composites (CSC) were prepared for the first time through a simple wet impregnation process and subsequent pyrolysis of low-value sugarcane by-products, namely molasses. These CSC materials demonstrate a distinct range of functionalities, which significantly differ from similar materials published in the literature. Importantly, the carbon-silica composites prepared at 800°C exhibited exceptional adsorption capacities for the azo-dye congo red (445 mg g$^{-1}$), due to the graphitic carbon coating and unique functionality including C-O-C within the porous structure. Congo red adsorption capacity of the highly mesoporous graphitic carbon-silica composites significantly exceeds that of commercial activated carbon and silica, these carbon-silica composites therefore represent an effective step towards the development of porous bio-derived adsorbent for remediation of dye wastewaters. Both the porous properties (surface area and pore size distribution) and the functionality of the carbon coating were dependent on the temperature of preparation. The sustainable synthetic methods employed led to a versatile material that inherited the mesoporosity characteristics from the parent silica, demonstrating mesoporous volumes greater than 90% (as calculated from the total pore volume). Adsorption on the 800°C prepared carbon-silica composites demonstrated an excellent fit with the Langmuir isotherm and the pseudo-first-order kinetic model.

# 1. Introduction

Dyes are used in a wide range of industrial sectors including paints, textiles, plastics, cosmetics and paper manufacture; however, an estimated 3% of all dyes produced are lost during the bleaching and colouring process, generating a large volume of industrial wastewaters [1,2]. Many organic dyes are considered allergens, skin irritants or even carcinogens [3] and thus are considered dangerous to the environment [4,5]. Dye removal processes can broadly be divided into two categories: chemical process and physical process. The chemical methods include coagulation-flocculation and chemical oxidation processes, while the physical methods include adsorption and membrane-separation processes [6]. Adsorption is an inexpensive and efficient method for the removal of dyes from wastewaters. The most commonly used adsorbents include activated carbon, peat, wood chips, fly ash, silica gel and natural clays [7].

Carbon-based materials have found use in a wide range of application owing to their unique three-dimensional structure, high surface area, thermal and chemical stability and decent conductivity [8]. Their unique characteristics have enabled a number of carbonaceous materials including carbon black, graphene, graphene oxide, carbon fibre and carbon nanotubes to be used in composites/hybrid materials in order to enhance their properties in various applications such as catalysis, nanoelectronics and sensors [9–13].

Porous carbonaceous materials derived from biomass are an important class of materials for the removal of organic pollutants from wastewaters. Activated carbons are one such group of adsorbent materials that frequently exhibit excellent adsorption capacities and some selectivity [14]. Typically such materials are highly microporous, hence can suffer from diffusion limitations [15]. This is especially problematic when adsorbing bulkier molecules such as dyes. In addition, recent work has demonstrated that activators significantly contribute to the cost of activated carbons and can potentially lead to significant quantities of acidic aqueous waste associated with activator removal from the carbon [16]. Mesoporous carbonaceous polysaccharide-based materials could be used to overcome such limitations and have some positive green credentials; however, production of such materials is laborious, resource intensive and costly [17,18]. By contrast, silicas typically exhibit low adsorption capacities but are mechanically stable, widely available, low cost and can possess hierarchical pore structures [15,19,20]. Combining the beneficial properties of carbons and silica would be advantageous for adsorption materials.

Carbon-silica composite (CSC) materials are polymer or chemical coatings that cover the surface of inorganic silica supports. The coated polymer layer over the silica materials is subsequently converted to carbonaceous layers through a pyrolysis process or acid treatments to yield the CSCs. They offer an effective method for the modification of silica, possessing several of the advantageous properties of carbons and silica. These novel materials have unique properties when compared to traditional activated carbons or templated mesoporous carbons. CSCs have been used in a wide range of applications including catalysis [21–23], drug delivery systems [24], energy storage [25] and adsorption [26–28]. To date, the use of such materials in adsorption applications has focused on the removal of heavy metals and also gaseous pollutants including $NH_3$, $SO_2$, $CO_2$, methane and ethane. Limited work has been reported on the use of CSC materials for the removal of organic molecules such as dyes.

CSCs have been prepared with a wide range of chemicals or polymers such as hydrocarbon gas ($C_3H_6$) [29], (poly)furfural alcohol (PFA), furfuryl alcohol (plus acid catalyst) [30] and dichloromethane as the carbon source [31]. However, many of these carbon sources are costly, potentially toxic or lead to toxic by-products during carbonization. Therefore, low-cost and non-toxic carbon sources, which are ideally bio-based, are needed for such materials. Recent work has demonstrated the use of pyrolysis oil from waste cellulose produced CSC materials with comparable porosity to the parent silicas; however, the composition of such carbon sources is heavily dependent on both the feedstock and process conditions [32] To date, little is known about the toxicity of such bio-oils, and although their use has led to enhanced aromaticity on heating to 800°C, the limited graphitic character was observed in the resulting CSC according to previously published XRD results [32]. Sucrose has been used as a safer more sustainable carbon source [33]; however, there are concerns over the use of high-quality and high-cost food products for the production of chemicals, materials and fuels. Low-cost beet sugar syrup can be used as an effective carbon source for the preparation of CSCs [34], thus highlighting the potential for the exploitation of low-value by-products of the sugar industry. With the production of sugarcane worldwide reaching 1.9 billion tons in 2016, a high volume of by-products/wastes including sugarcane tops and leaves from the field, bagasse and molasses from the sugar industry can be envisaged in the future [35]. These low-cost and highly available resources therefore present an attractive source of chemicals and materials production.

Blackstrap molasses is a lower value by-product residue of sugar production from sugarcane. It has limited use in confectionary manufacture but is used for low-value animal feed and as a carbon source for fermentation to ethanol and other chemicals [36]. Molasses contain up to 60% sugars, in addition to water (20%), inorganic ashes and other organic matter (gums, acids, lignin residues, amino acids and decomposition products) [37]. Molasses production in the European Union (EU 28) alone is estimated to increase to 3.5 million tons per year by 2027 [38].

Herein, a novel method for the production of previously unreported highly graphitic porous carbon coating on the surface of amorphous K60 silica gel has been developed from molasses. The synthetic methodology involves the use of 'green solvents' including water and ethanol. While, the carbonization of the molasses was investigated using thermogravimetric infrared spectroscopy (TG-IR) to determine the volatiles released during pyrolysis. By applying this simple wet impregnation method for molasses, followed by pyrolysis at different temperatures, highly graphitic carbon-silica composite materials with tunable properties, including surface porosity and functionality of the carbon coating were achieved. Molasses-based CSCs were characterized by CHN analysis, Fourier transform infrared spectroscopy (FT-IR), X-ray diffraction (XRD), X-ray photoelectron spectroscopy (XPS), Raman spectroscopy, nitrogen adsorption porosimetry, scanning electron microscopy (SEM) and transmission electron microscopy (TEM) and tested for the removal of dyes including azo-dye congo red (anionic dye) and also methylene blue (cationic dye). The experimental data were fitted to isothermal and kinetic models in order to investigate the adsorption mechanism. Bio-derived carbon-silica composites prepared from the low-value molasses are not only novel porous materials but this work also demonstrates that these CSC can be used as effective adsorbents for the removal of dyes.

# 2. Experimental

## 2.1. Materials

Molasses was obtained from Mitr Phol Thailand Industry (18% water content). Silica gel 60 (less than 0.063 mm) and ethanol were supplied by Merck. Methylene blue (C.I. 52 015), with the molecular formula $C_{16}H_{18}CIN_3S$, molecular weight of 319.85 g mol$^{-1}$ and the wavelength of maximum adsorption = 665 nm, was supplied by Ajax Finechem. Congo red (C.I. 22 120), with the molecular formula $C_{32}H_{22}N_6Na_2O_6S_2$, the molecular weight of 696.68 g mol$^{-1}$ and the wavelength of maximum adsorption = 498 nm, was obtained from Merck, USA.

## 2.2. Preparation of carbon-silica composites

The carbon-silica composites were prepared by dissolving molasses (40 ml) in water; then silica gel 60 (10 g) was added into the molasses solution (the molasses to silica ratio of 4/1 by weight was determined as the optimum ratio that had the highest incorporated molasses onto the silica, see electronic supplementary material). Ethanol (80 ml) was then added and the solution was stirred overnight. The supernatant liquid was removed by decantation followed by filtration prior to the drying in the oven. The obtained material was dried in an oven at 120°C for 2 h. The dried sample was carbonized at 400°C (CSC400), 600°C (CSC600) or 800°C (CSC800) under nitrogen, to yield the carbon-silica composites.

In order to investigate the phenomenon that occurs during pyrolysis process, the TG-IR was performed and the composition of the volatiles emitted in the course of the carbonization was examined. The samples were heated from 25°C to 800°C at 10°C min$^{-1}$ using a Netzsch 409 simultaneous thermal analyser equipped with a Bruker Equinox-55 FT instrument.

## 2.3. Characterization of carbon-silica composites

The elemental composition of the CSCs was analysed using a Perkin-Elmer PE-2400II CHN-analyser. The surface area and pore characteristics of the composites were determined via $N_2$ adsorption–desorption isotherms study using Autosorb 1-C instrument (Quantachrome) at −196°C. All samples were degassed under vacuum at 300°C for 3 h prior to the measurements. The specific surface areas were calculated by the multipoint Brunauer–Emmett–Teller (BET) method and pore size distributions were determined using the Barrett–Joyner–Halenda (BJH) method. The morphological characteristics were revealed using a scanning electron microscope (LEO model 1450VP) with an

accelerating voltage of 20 kV. Transmission electron microscopy (TEM) images were recorded using a TECNAI G2 (The Netherlands). ATR-FTIR spectra were recorded using a Bruker FTIR spectrometer (Tensor 27) with Opus 7.0 software measuring in the range of 4000–550 cm$^{-1}$. The crystalline structure of the composite was studied by powder X-ray diffraction (XRD) using a Philips X-ray diffractometer (PW3040, The Netherlands) with CuK$\alpha$ radiation ($\lambda = 0.15406$ nm). XPS measurements were performed on a Kratos Axis Ultra DLD spectrometer (Manchester, UK). Raman test was carried out using a Horiba XploRA plus Raman spectrometer with a 532 nm laser excitation source.

## 2.4. Adsorption studies

In order to determine the adsorption capacity of the CSCs, standard solutions of congo red and methylene blue were prepared at the concentrations of 20 and 6 mg l$^{-1}$, respectively. An equal concentration of each adsorbent at 10 mg l$^{-1}$ was used in batch experiments. In order to assure the equilibrium of the solid-solution mixture, the suspensions were stirred at 250 r.p.m. for 24 h at room temperature. Samples were taken by filtering through a syringe filter (0.45 μm, Millipore, USA). A UV-spectrometer was used to determine the concentration of congo red and methylene blue at the wavelengths of 498 and 665 nm, respectively. For the adsorption isotherm study, the concentrations of congo red used were in a range of 4–28 mg l$^{-1}$ and those of methylene blue ranged from 2 to 8 mg l$^{-1}$. Dye concentrations were selected to ensure a maximum absorbance of 1. The adsorption capacity ($q_e$) was determined using the following expression:

$$q_e = (C_0 - C_e)\frac{V}{m},$$

where $C_0$ is the initial concentration of the solution, $C_e$ is the concentration of the sample solution, $V$ is the volume of solution used (l) and $m$ is the mass of adsorbent (g). The effect of pH on the congo red removal by CSC was investigated by adjusting the pH value of dye solution in the pH range of 2–10 with dilute aqueous solutions of HCl or NaOH (0.1 M). For this study, 250 ml of the dye solution (20 mg l$^{-1}$) was mixed with 2.5 mg of CSC and the mixture was stirred for 24 h. This was then followed by filtration and the absorbance of the remaining dye solution was measured at 498 nm. The percentage of dye removal was calculated as follows:

$$\frac{C_i - C_e}{C_i} \times 100,$$

where $C_i$ and $C_e$ are the initial and final (equilibrium) concentrations of dye (mg l$^{-1}$), respectively. In order to investigate the mechanism of congo red and methylene blue adsorption, the experimental data of the CSCs were applied to the Langmuir and Freundlich isotherm equations. The kinetic study was also carried out using pseudo-first-order and pseudo-second-order equations.

# 3. Results and discussion

## 3.1. Synthesis of carbon-silica composites

Synthesis of CSCs was simply performed through the coating of molasses onto the silica using liquid antisolvent precipitation (LAP) process. LAP is based on the change of supersaturation caused by mixing the solution and the antisolvent [39]. The process requires two miscible solvents and the chemical must dissolve in the solvent but not in the antisolvent. In the case of molasses-based CSC, water in molasses is the solvent, while ethanol acts as an antisolvent. The addition of ethanol to the water-molasses suspension thus causes the precipitation of molasses onto silica particles, resulting in a desirable coating effect. The coated silica was subsequently dried and carbonized to achieve the composites. This process presents a facile and simple method for the production of CSCs from low-value by-product of the sugar industry. Furthermore, the use of water and a bio-derived solvent such as ethanol promotes the sustainability of CSCs production. Importantly, the synthesis method avoids the use of the catalyst (including H$_2$SO$_4$) which is frequently used in the conversion of sucrose to form the carbon layer of the CSC [31]. In order to investigate the carbonization of the CSC precursor, TG-IR experiment was conducted to examine the evolved gases. Figure 1 demonstrates the IR spectra taken at the maximum rates of molasses decomposition, which is consistent with CSC precursors. Both decomposition processes showed a similar evolution of gases including CO$_2$ and water. The

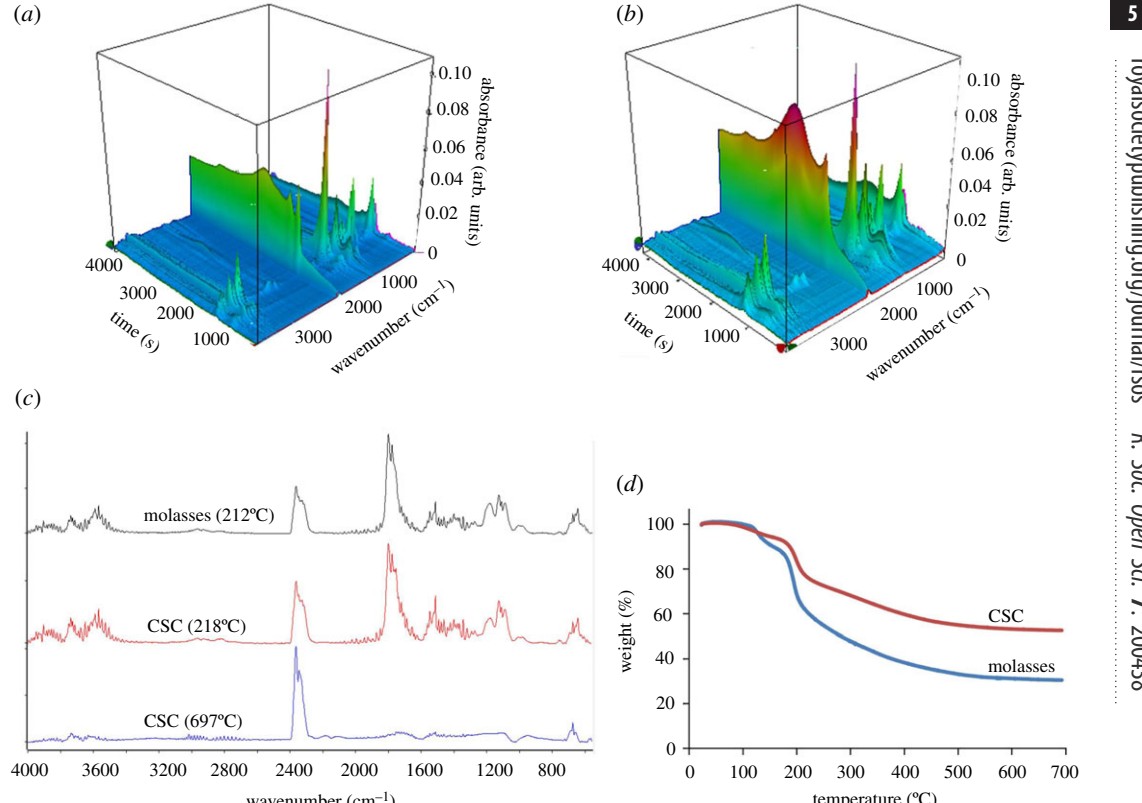

**Figure 1.** Three-dimensional plots of FT-IR spectra obtained during TG-IR experiments for (*a*) molasses and (*b*) CSC, (*c*) IR spectra taken at maximum rates of decomposition and (*d*) TG thermograms of molasses and CSC during TG-IR experiment.

pyrolysis of sucrose takes place in two stages: (i) dehydration occurred below 200°C and (ii) above this temperature gaseous volatiles are released from individual pyrolysis processes, where products such as 5-hydroxymethylfurfural and other dehydration products are produced [40]. This is also consistent with previous studies that investigated the pyrolysis behaviour of sucrose-containing biomass [41]. Both the decomposition of molasses and silica doped with molasses (CSC precursor) demonstrated a similar profile, with the majority of the decomposition occurring at 200–250°C. An increase in $CO_2$ production was observed at 300–400°C in the CSC material, as compared to molasses. High rate of decomposition was observed at the temperature near 700°C for CSC, as high intensity of $CO_2$ peak was noted. CSC materials have been prepared at 400°C (CSC400), 600°C (CSC600) and 800°C (CSC800). The CSC materials demonstrated a significant range of functionalities, materials prepared at 400°C were consistent with the sugar-derived nature of the carbon feedstock, while those materials prepared at temperatures above 600°C demonstrated greater graphitic character.

## 3.2. Properties of carbon-silica composites

Nitrogen adsorption–desorption isotherms of CSCs and the parent silica are demonstrated in figure 2. CSC400 and CSC800 exhibited type IV isotherms according to the IUPAC classification indicating mesoporosity [42]. CSC600 exhibited a combination of types I and IV isotherms, suggesting the existence of both micropores and mesopores in the structure. CSCs inherited mesoporosity characteristics from the parent silica, with the mesopore volumes greater than 90% for CSC800. This was less pronounced for the sample prepared at 600°C, due to a thicker carbonaceous layer as evidenced in lower average pore diameter (table 1) and increased microporous ratio.

Moreover, the elemental analysis revealed a higher percentage of carbon composition (table 2) of CSC600, confirming the superior carbon component within this material. This was further confirmed by TG analysis of CSC600, which demonstrated a 28% mass loss in the air, corresponding to the combustion of the carbon layer at 400°C (electronic supplementary material, figure S2). Micropores within the carbon coating prepared at 600°C may contribute to the higher microporous character of CSC600. BET surface areas of CSC materials decreased with increasing temperature of preparation.

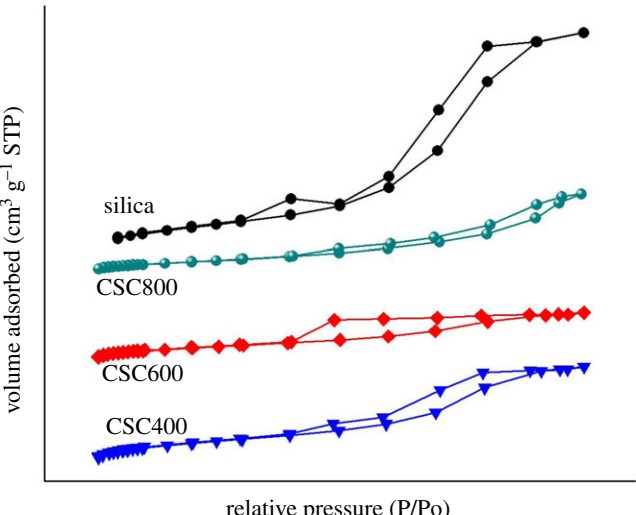

**Figure 2.** Adsorption/desorption isotherms of nitrogen onto CSC400, CSC600, CSC800 and silica.

**Table 1.** Nitrogen adsorption porosimetry for CSCs.

| sample | $S_{BET}$ ($m^2\,g^{-1}$) | $S_{micro}$ ($m^2\,g^{-1}$) | $S_{meso}$ ($m^2\,g^{-1}$) | $V_{total}$ ($m^3\,g^{-1}$) | $V_{micro}$ ($m^3\,g^{-1}$) | $V_{meso}$ ($m^3\,g^{-1}$) | $V_{micro}/V_{total}$ % | $V_{meso}/V_{total}$ % | $D_p$ nm |
|---|---|---|---|---|---|---|---|---|---|
| CSC400 | 265 | 57 | 208 | 0.35 | 0.026 | 0.328 | 7.40 | 92.59 | 5.35 |
| CSC600 | 225 | 92 | 133 | 0.21 | 0.045 | 0.168 | 21.07 | 78.87 | 3.79 |
| CSC800 | 135 | 16 | 119 | 0.27 | 0.007 | 0.260 | 2.83 | 97.16 | 7.94 |
| silica | 308 | 16 | 292 | 0.73 | 0.007 | 0.732 | 0.96 | 99.05 | 9.60 |

**Table 2.** Elemental composition of CSCs.

| sample | %C | %H | %N |
|---|---|---|---|
| CSC400 | 15.67 | 1.48 | 0.14 |
| CSC600 | 20.79 | 1.15 | 0.17 |
| CSC800 | 13.64 | 0.48 | 0.04 |

The surface areas of the composites were reduced after the introduction of the carbon coating to the silica. A similar trend was also observed in the CSCs prepared from MCM-41 and polyfurfuryl alcohol, nonetheless in this case, the composites retained no mesoporosity of the parent silica as observed in the present study [27].

The FT-IR spectra of CSCs (figure 3) showed characteristic bands of silica including ones at 3500, 1060 and 800 $cm^{-1}$ which were attributed to O-H stretching, asymmetry Si–O–Si bond stretching and $SiO_4$ tetrahedron ring, respectively [32,43]. CSC400 and CSC600 spectra exhibited a band at 1620 $cm^{-1}$, representing the stretching of C=C. Notably, the sample prepared at 600°C showed a band at 1456 $cm^{-1}$ which was assigned to C-H from methylene group, while the material that was carbonized at 800°C showed a band at 1582 $cm^{-1}$ which is characteristic of C=C aromatic skeletal vibrations. This suggested that the aromatic structures were formed during high-temperature carbonization and this was consistent with XRD results within this study.

Figure 3*b* shows the XRD patterns of the CSCs. A peak at 22° which is characteristic of the cristobalite structure of silica was observed for all spectra [34,44]. The intensity of this broad peak was less intense in CSC600, confirming the higher carbon to silica ratio as seen in TGA and CHN analysis. Interestingly, two diffraction peaks of graphitic carbon at 25° and 43° which correspond to (002) and (100) or (101) planes of crystalline hexagonal graphite lattice, respectively, were clearly observed [45–48]. This confirms the FT-IR

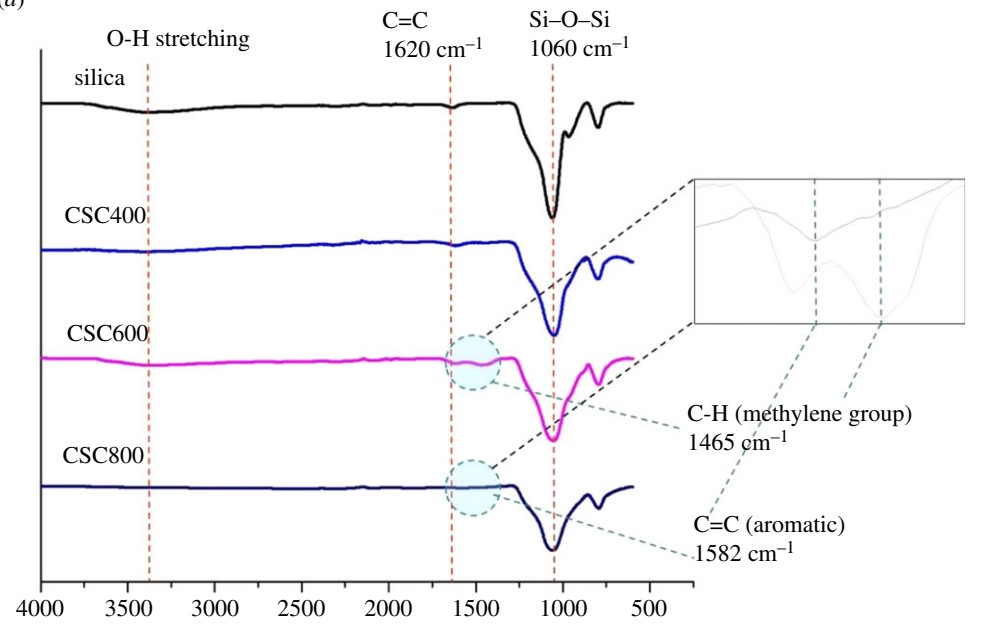

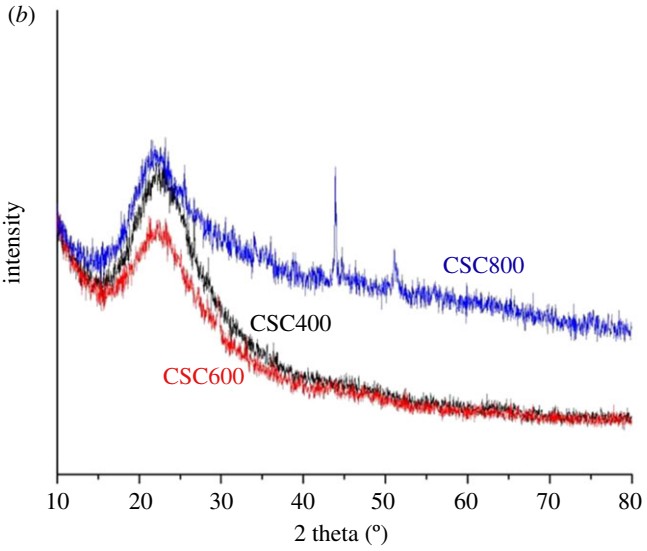

**Figure 3.** (a) FT-IR spectra of CSC400, CSC600, CSC800 and silica and (b) XRD patterns of CSC400, CSC600, CSC800.

results that the carbonization of this composite at 800°C gives rise to the graphitic carbon structure, which is in accordance with previous reports of CSC from sucrose [21].

XPS was used to determine the elemental content on the CSCs surface (figure 4 and table 3). XPS demonstrates that % C content on the surface of the CSCs slightly increases with higher temperatures of preparation. In comparison, % O stays consistent up to 600°C and then drops at 800°C. The carbon to oxygen ratio (C/O) on the surface of the CSC materials increased from 0.51 to 0.64 to 0.77 for CSC400, CSC600 and CSC800, respectively. Per cent Si content is lowest at 600°C and the carbon to silicon ratio (C/Si) as determined by XPS was 1.26, 2.55 and 1.75 for CSC400, CSC600 and CSC800, respectively. This thus confirms that the greatest coverage of carbon on the surface occurred at a temperature of 600°C.

XPS C1S demonstrates contributions from C=C, C–C, C–O, C=O and C–O–Si groups on the surface of CSC at 284.2, 284.9, 286.2, 287.2 and 288.8 eV binding energies, respectively [32,49]. XPS results also indicate that CSC400 carbon surface is still dominated by aliphatic carbon functionalities (C-C sp3, 284.9 eV). As the temperature of CSC preparation is increased to 600°C, a greater proportion of C=O (287.2 eV) and C-O (286.2 eV) can be observed in the XPS. CSC600 demonstrates a unique balance between aliphatic carbon and carbonyl-containing functionalities on the surface of the CSC. By contrast,

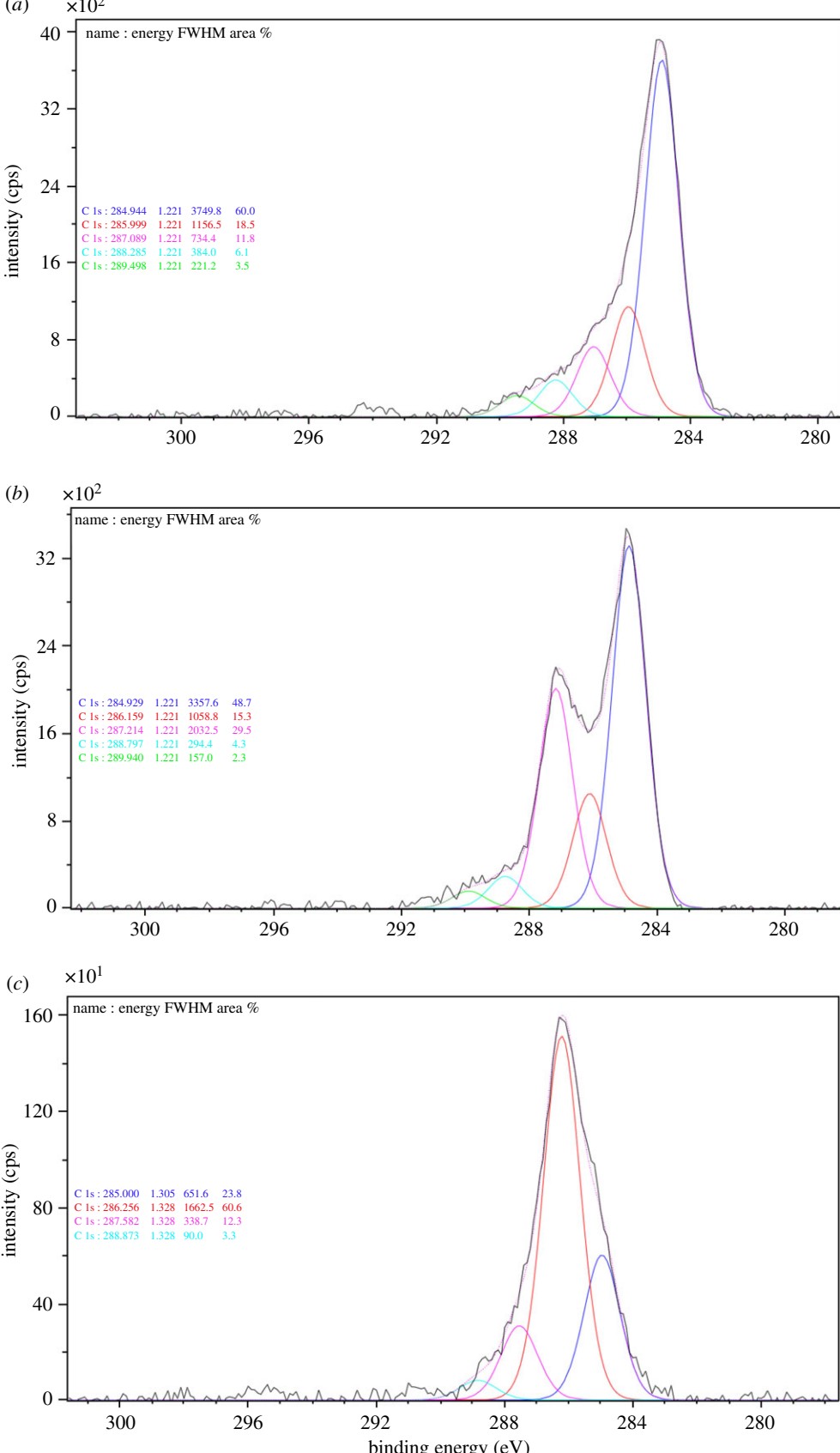

**Figure 4.** C1s XPS spectra of (*a*) CSC400, (*b*) CSC600 and (*c*) CSC800.

**Table 3.** % atomic content of CSC400, CSC600 & CSC800 from C1 s XPS.

| sample | % atomic content | | | |
| --- | --- | --- | --- | --- |
| | C | Si | O | other |
| CSC400 | 26.77 | 21.19 | 52.04 | 0.00 |
| CSC600 | 33.63 | 13.19 | 52.78 | 0.94 |
| CSC800 | 34.73 | 19.86 | 45.17 | 0.24 |

at 800°C, the contribution of aliphatic carbon is greatly diminished and the material is dominated by C-O (C-O-C, C-O-H). In all cases, oxygen is the bridge between the carbon layer and the silica, leading to the observed C–O–Si bonds (288.8 eV), and this is consistent with observations of the CSC materials [49]. XPS demonstrated that the proportion of C=C $sp^2$ (284.2 eV) on the surface of all molasses-based CSC materials was low and this is in contrast with CSC materials produced from bio-oil [49]. XPS for O1S and Si2P of CSC400, CSC600 and CSC800 are available in the electronic supplementary material.

Raman spectra (electronic supplementary material, figure S5) show a silicon band at around 470 cm$^{-1}$ [50]. A defect band (D band) of disordered graphene sheets was observed at 1375 cm$^{-1}$; however, the graphitic band (G band) that is usually seen around 1600 cm$^{-1}$ was not pronounced [51]. Various bands between 1690 and 1820 noted in CSCs spectra were attributed to M band, which only appears in AB-stacked bilayer graphene (BLG) or few-layer graphene (FLG) caused by an intravalley double resonance scattering process [52]. XPS data is in contrast with FT-IR, XRD and Raman that indicate the likelihood of graphitic-type carbon structures within the internal network of the CSC. It is possible that these CSC materials have a proportion of graphene or graphitic layers within the internal structure (with limited amounts on the external surface); however, these internal graphitic-like structures are likely to demonstrate disorder [53].

Figure 5 displays SEM and TEM images of the CSCs and silica. The textural properties of CSCs as well as the porous characteristics are consistent with the parent silica, where many of these properties were retained in all the composites. The darker areas on the TEM images correspond to the silica and the lighter areas surrounded the silica particles are the carbon coating [29]. According to figure 5e–h, a uniform coating of low-density carbonaceous layer is observed in all CSC materials. This is further supported by scanning electron microscopy -energy-dispersive X-ray spectroscopy (SEM-EDS) data which demonstrates a uniform coverage of carbon (figure 6 and electronic supplementary material).

## 3.3. Adsorption isotherms

Analysis of adsorption isotherms is essential for understanding the surface interaction between the adsorbent and the adsorbate. The isotherm data provide important parameters for the design of an adsorption system including the prediction of the adsorption capacity [54–56]. Langmuir and Freundlich isothermal models were applied to investigate the adsorption process of congo red and methylene blue. Langmuir isotherm assumes the homogeneous structure of adsorbent and no interaction between adsorbate molecules. The model describes the formation of monolayer adsorption onto energetically uniform sorption sites [57]. The linear form of Langmuir isotherm can be expressed as follows:

$$\frac{C_e}{q_e} = \frac{1}{K_L} + \frac{a_L}{K_L} C_e$$

and

$$Q_0 = \frac{K_L}{a_L},$$

where $C_e$ is the concentration of the adsorbate in the solution at equilibrium (mg l$^{-1}$), $q_e$ is the adsorption capacity at equilibrium, $Q_0$ is the monolayer adsorption capacity of the adsorbent (mg g$^{-1}$) and $a_L$ and $K_L$ are Langmuir adsorption constants [10]. The values of $a_L$ and $K_L$ are determined from the slope and intercept of the plot of $C_e/q_e$ versus $C_e$ [17].

The Freundlich equation describes adsorption on heterogeneous surfaces which does not restrict to the monolayer coverage. The linear form of Freundlich isotherm is given by the following equation [58]:

$$\ln q_e = \ln k_f + \frac{1}{n} \ln C_e,$$

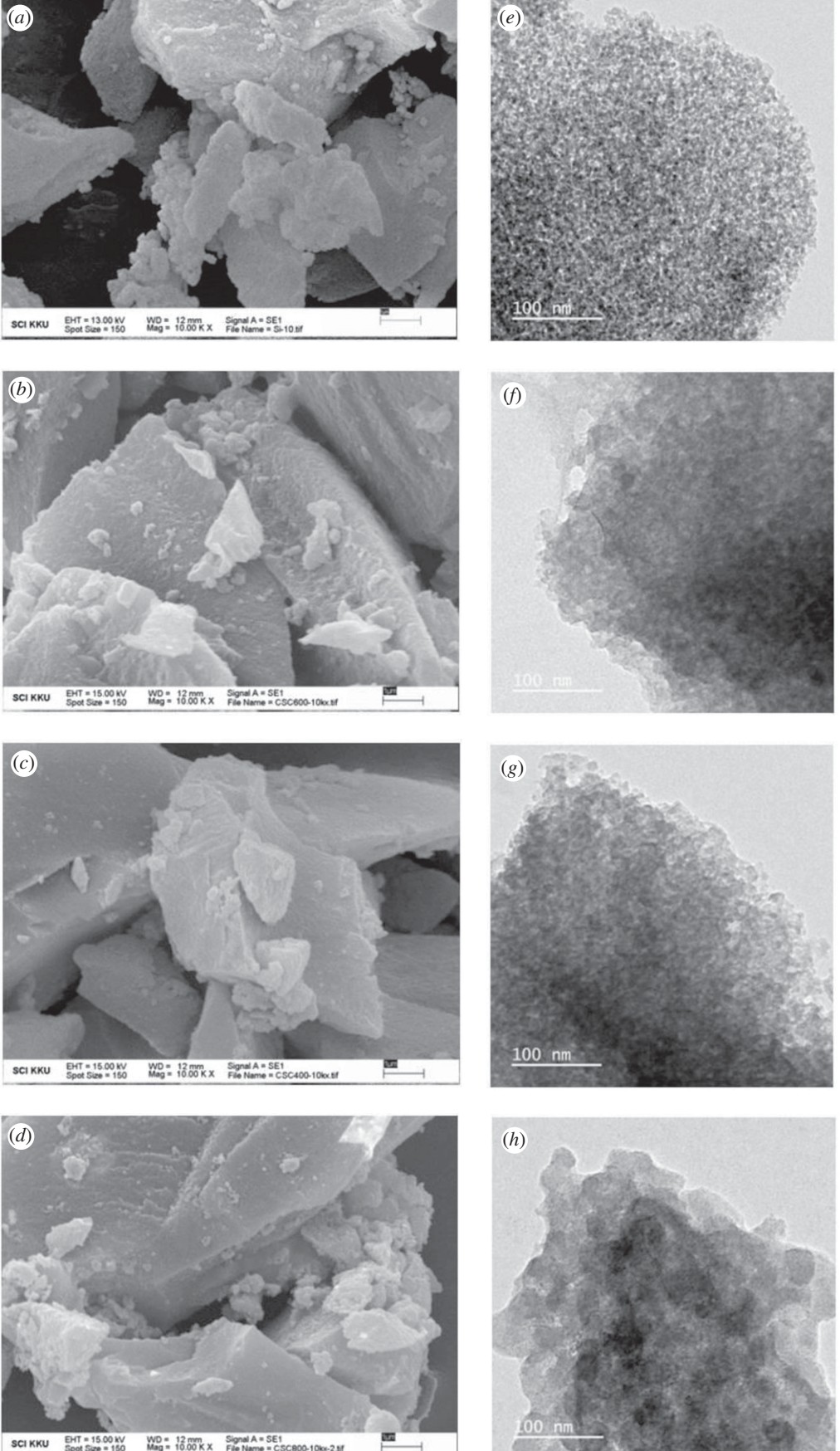

**Figure 5.** SEM images (left column) and TEM images (right column) of silica (*a,e*), CSC400 (*b,f*), CSC600 (*c,g*) and CSC800 (*d,h*).

(a) electron image 1

(b) EDS layered image 1

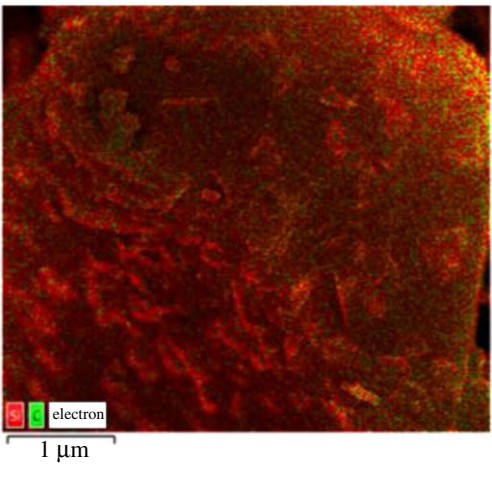

(c) C Kα1_2

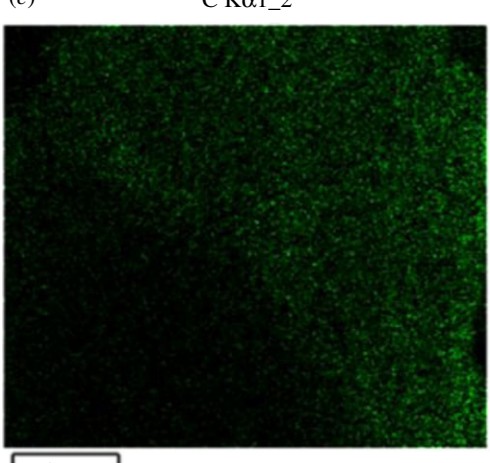

(d) Si Kα1

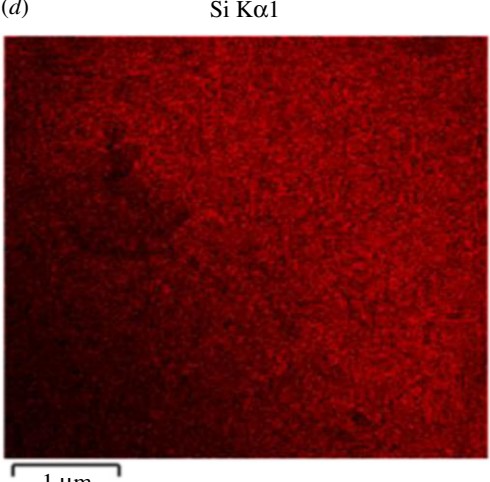

**Figure 6.** SEM-EDS mapping of carbon and silica in CSC600.

where $k_f$ is the Freundlich constant ($l\,g^{-1}$) which is indicative of adsorption capacity of the adsorbent. $1/n$ is the heterogeneity factor and represents the intensity of the adsorption process. An $n$ that is equal to unity suggests linear adsorption, while $n$ below unity predicts a chemical adsorption process. An $n$ that is greater than unity indicates favourable physical adsorption [56]. The Freundlich constants can be calculated from the slope and the intercept of a plot of log $q_e$ versus log $C_e$ [59].

Regarding the adsorption of congo red, the adsorption isotherms are presented in figure 7a,b and the calculated parameters are summarized in table 4. The isothermal data fitted well with both Langmuir and Freundlich isotherms, indicating that both monolayer and multilayer adsorption might occur. It was noted that the experimental adsorption capacity of CSC600 was at $291\,mg\,g^{-1}$ (table 6), which was close to the theoretical monolayer adsorption capacity ($290\,mg\,g^{-1}$). The highest monolayer adsorption capacity was obtained from CSC800 at $390\,mg\,g^{-1}$, while the experimental adsorption capacity for this material was $Q_{exp} = 455\,mg\,g^{-1}$, which was attributed to the highly aromatic carbon surface of the adsorbent. By contrast, CSC400 demonstrates experimental adsorption of $Q_{exp} = 381\,mg\,g^{-1}$, which far exceeded the predicted $269\,mg\,g^{-1}$ monolayer adsorption capacity. This thus demonstrates that CSC400 is likely to undertake multilayer adsorption. The $n$ values suggested that the adsorption was more favourable in CSC600 and CSC800 compared to that of CSC400, which was potentially due to the enhanced graphitic nature and aromaticity of the former. Typical mechanisms of the adsorption process of congo red include electrostatic interaction, hydrogen bonding and pi-pi interactions [60,61]. The latter involves the interaction between the delocalized pi-electrons on the basal planes of carbon adsorbent and the free electrons of the aromatic rings of congo red molecules [62]. The diagram representing such a process is shown in figure 8. Importantly, the unique range of functional groups present in carbon matrix of molasses-based CSC such as

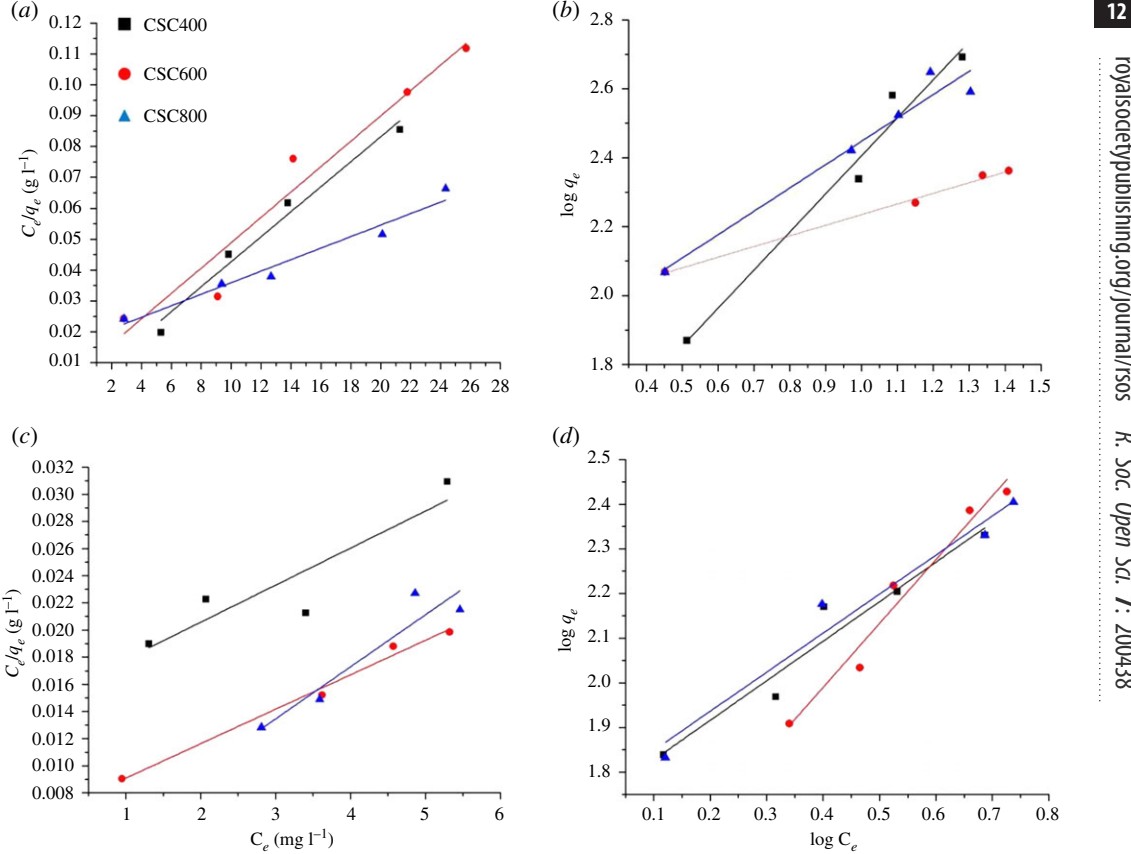

**Figure 7.** Adsorption isotherms for adsorption of congo red (*a,b*) and methylene blue (*c,d*) on CSCs according to Langmuir (*a,c*) and Freundlich (*b,d*) models.

**Table 4.** Isotherm parameters for congo red adsorption on CSCs.

| isotherm | isotherm parameter | CSC400 | CSC600 | CSC800 |
|---|---|---|---|---|
| Langmuir | $a_L$ (l mg$^{-1}$) | 1.80 | 0.52 | 0.10 |
| | $Q_0$ (mg g$^{-1}$) | 269 | 290 | 390 |
| | $R^2$ | 0.9686 | 0.9305 | 0.9547 |
| Freundlich | $k_f$ | 3.67 | 6.86 | 5.87 |
| | n | 0.90 | 3.23 | 1.47 |
| | $R^2$ | 0.9609 | 0.9937 | 0.9431 |

C-O-C, C–OH, C=O and COO- could also play a vital role in the adsorption mechanism of the dye [63]. The proportions of these functionalities clearly demonstrate significant differences over other CSC materials in the literature [49]. This is especially true for CSC600 and CSC800, where XPS demonstrates the formation of C=O at 600°C, while C-O-C functional groups are dominated at 800°C.

In the case of methylene blue, it was noted that silica might play a role in the cationic dye adsorption as it was reported that mesoporous silica could selectively adsorb cationic molecules. This is due to the electrochemical interaction between the cationic molecules and the negative charges on the silica surfaces. However, the effect of silica adsorption in the composites should be minimal as the SEM-EDS images showed uniform coverage of carbon onto silica [64]. Although silica may not be directly involved in the adsorption process, the mesoporous structure it provides is likely to enable the rapid diffusion of dye molecules into the CSC framework. Adsorption isotherms of methylene blue are shown in figure 7*c* for Langmuir isotherm and figure 7*d* for Freundlich model. Table 5 exhibits the values of calculated

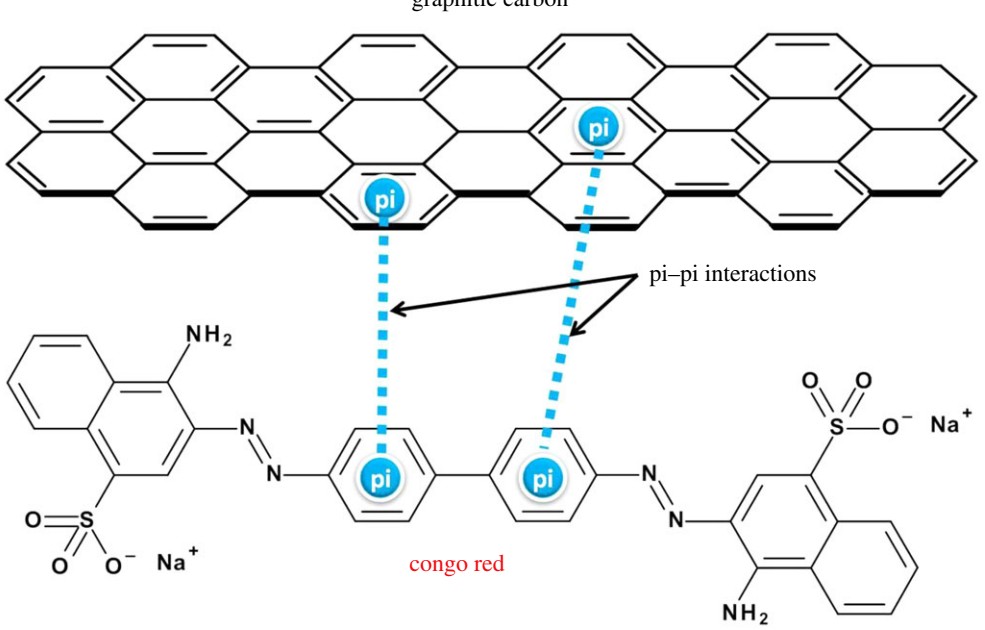

**Figure 8.** Pi–pi interactions between graphitic carbon materials and congo red.

**Table 5.** Isotherm parameters for methylene blue adsorption on CSCs.

| isotherm | isotherm parameter | CSC400 | CSC600 | CSC800 |
|---|---|---|---|---|
| Langmuir | $a_L$ (l mg$^{-1}$) | 0.18 | 0.38 | 1.96 |
| | $Q_0$ (mg g$^{-1}$) | 171 | 268 | 254 |
| | $R^2$ | 0.7423 | 0.9840 | 0.8488 |
| Freundlich | $k_f$ | 5.69 | 3.13 | 5.81 |
| | n | 1.12 | 0.87 | 0.56 |
| | $R^2$ | 0.9256 | 0.9582 | 0.9487 |

parameters. Freundlich isotherm provided a better fit for CSC400 and CSC800, while the data from CSC600 fitted better with Langmuir isotherm. This was attributed to a greater number of mesopores within CSC400 and CSC800, promoting more than one single-layer coverage on the heterogeneous surface.

The monolayer adsorption capacities of these CSC materials for methylene blue were lower than ones for congo red. This may be due to limited interactions between the dye and adsorption sites, such as the pi–pi interaction and electrostatic bonding. The experimental adsorption capacities of both CSC600 and CSC800 were very close to the theoretical monolayer adsorption capacities, while CSC400 showed much higher experimental value, indicating that the adsorption in the latter should consist of both monolayer and multilayer adsorption. CSC600 showed slightly higher monolayer adsorption capacity (268 mg g$^{-1}$) than that of CSC800 (254 mg g$^{-1}$). In adsorption, macropores and mesopores play an important role as the transport arteries for the adsorbate, making the internal parts of the adsorbent readily accessible. However, for larger molecules such as some dyes or biological molecules, mesopores may be the active sorption sites. Nonetheless, micropores represent high adsorption potential due to the opposite wall adsorption force in their entire volume and high surface energies [65]. In this case, the combination of micropores and mesopores in CSC600 should promote the adsorption of methylene blue, leading to higher monolayer adsorption capacity.

## 3.4. Adsorption kinetics

Examination of the kinetics and diffusion process during the adsorption, pseudo-first- and pseudo-second-order kinetics models were studied. Pseudo-first-order equation describes the adsorption of

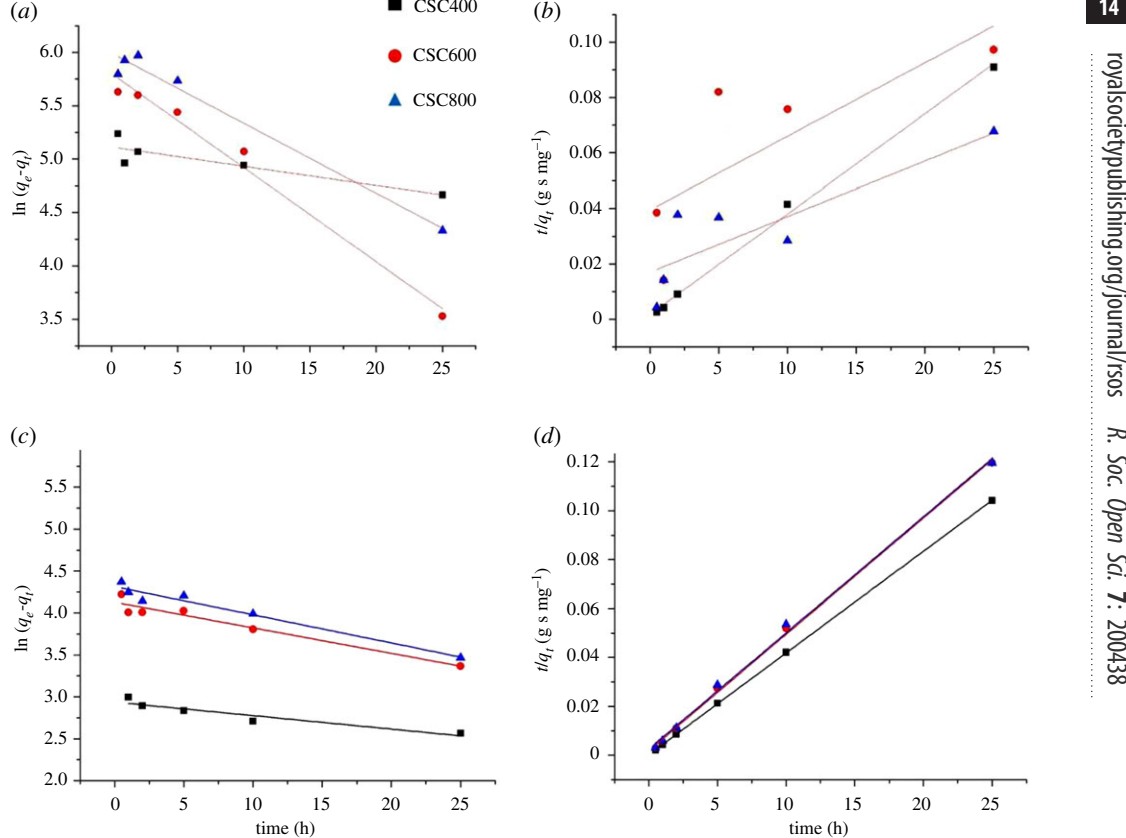

**Figure 9.** Adsorption kinetic for adsorption of congo red (*a,b*) and methylene blue (*c,d*) on CSCs according to pseudo-first-order (*a,c*) and pseudo-second-order (*b,d*) models.

adsorbate from an aqueous solution, assuming that the rate of change of solute uptake with time is directly proportional to the difference in saturation concentration and the amount of solid uptake with time. The pseudo-first-order kinetic equation is formulated as follows:

$$\log(q_e - q_t) = \log q_e - \left(\frac{k_1}{2.303}\right)t,$$

where $q_e$ and $q_t$ are the adsorption capacity at equilibrium (mg g$^{-1}$) and at time $t$ (min), respectively, and $k_1$ is the pseudo-first-order rate constant (min$^{-1}$).

The pseudo-second-order kinetic model describes chemisorption as well as cation exchange reactions [66]. The pseudo-second-order kinetic equation is represented by the following equation:

$$\frac{t}{q_t} = \frac{1}{k_2 q_e^2} + \frac{t}{q_e},$$

where $k_2$ is the pseudo-second-order rate constant (g mg$^{-1}$ min$^{-1}$).

Figure 9*a,b* presents the plots of the pseudo-first-order and pseudo-second-order kinetics of congo red adsorption on CSCs, and the kinetic parameters are exhibited in table 6. The obtained results indicated that the adsorption process of CSC400 followed a second-order kinetic model while ones of CSC600 and CSC800 fitted better with the first-order equation. This is confirmed by the fact that the experimental adsorption capacities of CSC600 and CSC800 were closer to $q_e^1$ when the value of CSC400 was closer to $q_e^2$. Concerning methylene blue adsorption, as can be seen from figure 9*c,d*, the experimental data exhibited an ideal fit to the pseudo-second-order kinetic model with extremely high correlation coefficient ($R^2 > 0.99$) (table 7). Also the $q_e^2$ values calculated from pseudo-second-order equation were more consistent with the $q_e$ value from the experimental results. Therefore, the adsorption process of all CSCs for methylene blue is well represented by a pseudo-second-order kinetic model. This indicates that the rate of the direct adsorption/desorption of methylene blue on the surface controls the overall sorption kinetics of the process.

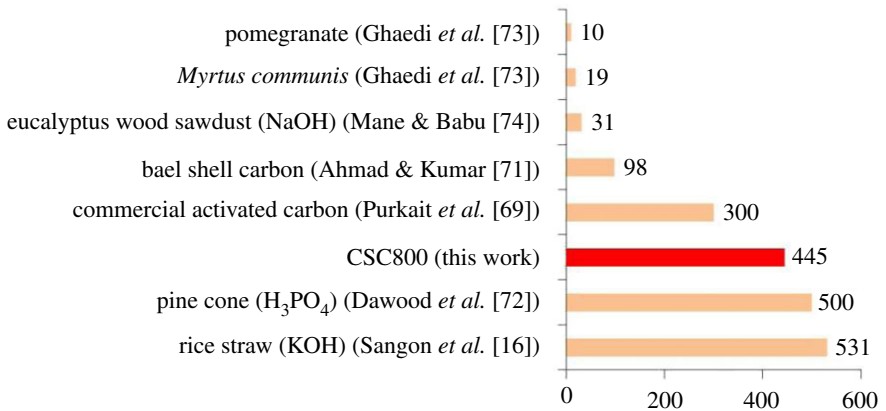

**Figure 10.** Literature reports on adsorption capacity of congo red from different biomass-derived adsorbents (activating agent in parenthesis).

**Table 6.** Adsorption parameters of kinetic study for the adsorption of congo red on CSCs.

| materials | $q_e$ (exp) (mg g$^{-1}$) | pseudo-first order | | | pseudo-second order | | |
|---|---|---|---|---|---|---|---|
| | | $q_{e1}$ | $k_1$ | $R^2$ | $q_{e2}$ | $k_2$ | $R^2$ |
| CSC400 | 381 | 166 | 0.0179 | 0.7330 | 277 | $7.62 \times 10^{-3}$ | 0.9961 |
| CSC600 | 291 | 293 | 0.0827 | 0.9555 | 384 | $1.71 \times 10^{-4}$ | 0.4833 |
| CSC800 | 445 | 399 | 0.0656 | 0.9690 | 500 | $2.35 \times 10^{-4}$ | 0.6608 |

**Table 7.** Adsorption parameters of kinetic study for the adsorption of methylene blue on CSCs.

| materials | $q_e$ (exp) (mg g$^{-1}$) | pseudo-first order | | | pseudo-second order | | |
|---|---|---|---|---|---|---|---|
| | | $q_{e1}$ | $k_1$ | $R^2$ | $q_{e2}$ | $k_2$ | $R^2$ |
| CSC400 | 294 | 19 | 0.016 | 0.8575 | 240 | 0.0774 | 0.9999 |
| CSC600 | 268 | 62 | 0.030 | 0.9255 | 210 | 0.0125 | 0.9983 |
| CSC800 | 254 | 75 | 0.034 | 0.9501 | 211 | 0.0087 | 0.9966 |

The effect of pH on the efficiency of the CSC adsorbents on the adsorption of congo red was also studied at the pH values of 2–10. It was observed that the removal of congo red is more efficient in acidic solution in which congo red dissociates to sulfonic group ($-SO_3^-$) (electronic supplementary material, figure S13) [67]. The carbonaceous surface of CSC is positive in acidic medium, hence the induced electrostatic interactions between the dye and the adsorbent favoured the adsorption of the dye onto the CSCs [68]. Similar results have been obtained for the adsorption of congo red onto activated carbons [68,69]. CSC materials exhibited a high percentage removal of congo red in acidic solution at 97–98% and the similar value was also observed for graphene quantum dots whose preparation would require more complex method [70]. It was noted that the CSCs prepared at different temperatures showed no significant differences in the influence of pH on dye removal.

The CSCs showed adsorption capacities for methylene blue in the region of 250–300 mg g$^{-1}$ which are comparable with those of commercial activated carbon [69]. However, for the adsorption of congo red, the adsorption capacity of CSC800 significantly exceeds that of commercial activated carbon (figure 10) [71]. In addition all the CSC materials demonstrated adsorption capacities that exceeded the parent silica ($89 \pm 9$ mg g$^{-1}$ for congo red and $135 \pm 2$ mg g$^{-1}$ for methylene blue). The composite sits among one of the most effective bio-derived adsorbents alongside activated carbons from rice straw and pine cone [72–74]. Importantly, CSC materials do not require the need for activators in their preparation, which have been reported to be the most expensive component of activated carbon synthesis [16]. Another distinct advantage of these materials is that at the end of life the carbon

coating can be combusted off, thereby generating energy and yielding a silica support framework which can be re-used, thus further enhancing the green credentials of these materials.

## 4. Conclusion

CSC materials have been synthesized with a range of distinct functionalities. These materials differ significantly from similar materials previously published in the literature. FT-IR, XRD and Raman indicated the potential formation of graphitic carbon layers on the internal surface of K60 silica gel through a simple, green and facile wet impregnation of molasses followed by pyrolysis. XPS analysis indicated a clear change in the surface functionalities from aliphatic carbon at 400°C, to a greater proportion of C=O and C-O at 600°C and finally leading to material at 800°C that was dominated by C-O (C-O-C, C-O-H). The synthesis method used low-value feedstocks, green solvents and importantly avoided the use of catalysts or activators. The resulting CSC materials were demonstrated as effective adsorbents for dye removal from aqueous media. Variation of pyrolysis temperature led to tunable materials with a significant range of functionalities. The FT-IR and XRD results confirmed the formation of graphitic materials at temperatures above 600°C. Nitrogen adsorption porosimetry and morphological study using SEM and TEM demonstrated the presence of the carbon coating on the silica support. The CSC material prepared at 800°C exhibited adsorption capacities in excess of 400 mg g$^{-1}$ for the azo-dye congo red, due to its graphitic nature and range of C-O-C functionalities (as determined through XPS). CSC800 demonstrated a good fit with Langmuir isotherm and the pseudo-first-order kinetic model. Our future work will focus on the reusability of these materials including cycle tests, the burn-off of the carbon coating at the end of life to generate energy and the reuse of the silica support. This work demonstrates that wastes can be used to overcome the challenges associated with using a pure feedstock for the development of CSC materials. Waste utilization can present new opportunities for the development of cost-effective advanced functionalized materials. Bio-derived carbon-silica composites prepared from the low-value molasses demonstrate promise as cost-effective adsorbents for water remediation and may find uses in a wide variety of other applications from catalysis to energy storage.

Data accessibility. The datasets supporting this article have been uploaded as part of the electronic supplementary material.
Authors' contribution. I.J. carried out the molecular laboratory work, collected data, participated in data analysis and participated in drafting the manuscript; A.J.H. participated in conceiving the study, data analysis and drafted the manuscript; Y.N. collected porosimetry data and carried out laboratory-based pyrolysis experiments; S.Y. participated in interpreting the data and the preparation of the manuscript; N.S. conceived the study, designed the study, coordinated the study, carried out data analysis, interpreted the results and helped draft the manuscript. All authors gave final approval for publication.
Competing interests. The authors declare no competing interests.
Funding. The work was financially supported by the Thailand Research Fund grant for New scholars, Office of the Higher Education Commission, and Khon Kaen University (grant no. MRG6280199), studentship funding of the Center of Excellence for Innovation in Chemistry (Perch-CIC) and the Newton Mobility Grant (Royal Society and OHEC, Thailand) (grant no. NI150336) for researcher exchange.
Acknowledgements. The authors graciously thank Mitr Phol Thailand Industry for supplying the molasses feedstock. Special thanks go to Mr. Paul Elliott (University of York) for his assistance in conducting TG-IR experiments. N.S. would like to acknowledge the financial support of the Thailand Research Fund, Office of the Higher Education Commission, and Khon Kaen University (grant no. MRG6280199). The authors would like to thank the Center of Excellence for Innovation in Chemistry (Perch-CIC) for studentship funding. The authors acknowledge the assistance and collaboration of the Green Chemistry Centre of Excellence at the University of York, through the Newton Mobility Grant (Royal Society and OHEC, Thailand) (grant no. NI150336).

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
