## [Reviewer comments · Royal Society Open Science]

Review History

RSOS-200438.R0 (Original submission)

Review form: Reviewer 1

Is the manuscript scientifically sound in its present form?

Yes

Are the interpretations and conclusions justified by the results?

Yes

Is the language acceptable?

Yes

Do you have any ethical concerns with this paper?

No

Have you any concerns about statistical analyses in this paper?

No

Recommendation?

Accept with minor revision (please list in comments)

Comments to the Author(s)

The article is technically good. I suggest to put more attention to Fig. 7 and discuss in detail the mechanism. In addition to obvious "electrostatic interaction, hydrogen bonding and pi-pi interactions", other interactions could be between O- and N-doped atoms, which frequently present in carbon matrix. The role of silica in adsorption process should be emphasized. Main recent books in dye removal by nanocomposites, hybrids and nanocomposites in general should be added and cited.

Review form: Reviewer 2

Is the manuscript scientifically sound in its present form?

Yes

Are the interpretations and conclusions justified by the results?

Yes

Is the language acceptable?

Yes

Do you have any ethical concerns with this paper?

No

Have you any concerns about statistical analyses in this paper?

No

Recommendation?

Accept with minor revision (please list in comments)

Comments to the Author(s)

This is a very interesting paper about a very nice approach to composite materials which would be expected to have very positive properties. It uses a large volume waste material from sugar production as a carbon source, which is an intelligent way to produce the materials, improving the literature routes to other similar materials.

There are a few fairly minor matters I would like the authors to address before publication:

The CSC600 material seems unusual – I would expect the C content to decrease as T increases.

Thermal analysis traces of mass loss suggest that it should be slightly lower.

Thermal treatment (TGIR data) – clearly also some organics evolved at low (200-250oC) temperatures. Do the authors have any insights into what these may be? Presumably

furaldehydes and similar dehydration products, as the traces seem very similar to molasses itself.

Abstract – mesopore volumes greater than 90% (presumably of the total pore volume?)

Experimental – how much water was added to the molasses? What is the water content of the molasses?

There must be a significant amount of supernatant liquid after mixing – is this removed before oven treatment?

XRD graphite at 52 and 43 (not 25 and 43)?

Review form: Reviewer 3

Is the manuscript scientifically sound in its present form?

Yes

Are the interpretations and conclusions justified by the results?

Yes

Is the language acceptable?

Yes

Do you have any ethical concerns with this paper?

No

Have you any concerns about statistical analyses in this paper?

No

Recommendation?

Major revision is needed (please make suggestions in comments)

Comments to the Author(s)

This manuscript is needed a major revision and further review. The comments are given below.

Limitation of the work, novelty and contributions should be highlighted more.

Following references should be included in the introduction part for more readable, about various carbon nanostructures and its structure-properties. *Int. J. Hydrogen Energy* 45 (2020) 7716-7740; *Carbon* 149 (2019) 693-721; *Synthetic Metals* 159 (2009) 1934-1939; *Microchemical Journal* 149 (2019) Article 103985; *Small methods* 2 (2018) Article 1800050; *Scripta Materialia* 58 (2008) 1010-1013; *J. Environ. Manage.* 238 (2019) 25-40; *ChemistrySelect* 4 (2019) 5322-5337; *Int. J. Hydrogen Energy* 44 (2019) 10453-10472; *J. Environ. Manage.* 254 (2020) Article 109747; *Chemical Eng. Journal* 309 (2017), 151-158; *J. Environ. Manage.* 250 (2019) Article 109486; *Chemistry - An Asian J.* 13 (2018) 3561-3574.

Please compare and discuss the differences between diffractions in Figure 3B.

The growth mechanism for the formation of CSCs should be explained in more detail based on the chemistry concept.

characterization data is needed for this work. It includes HRTEM images, SAED, EDS, XPS and Raman data.

DTG curves should be included in TGA curves to see accurate solvent removal and thermal degradation temperatures. Please tabulate the different thermogravimetric parameters (Tonset, Td, Tg, ΔT and % char).

Theoretical maximum adsorption capacity should also be calculated and compare with experimental maximum adsorption capacity.

What about the cycle tests of adsorption process for congo redy removal? What is the effect of pH on adsorption performance?

The authors should update the following references in the results part, about the different techniques used for removal of chemical pollutants: *Chem Eng. J* 385 (2020) Article 123787; *Small methods* 2 (2018) Article 1800050; *Separation and Purification Tech.* 210 (2019) 850-866; *J. Environ. Manage.* 265 (2020) 110504; *J. Nanosci. and Nanotech.* 10 (2010) 7951-7957; *Chemosphere* 239 (2020) 124766.

What would be the effect of different conc. of molasses and higher carbonization temperature (>800 degC) on the adsorbent performance.

In the conclusion part, the authors should mention about the future direction of the work.

Decision letter (RSOS-200438.R0)

Dear Dr Hunt:

Title: Highly graphitic mesoporous carbon-silica composites from low value sugarcane by-products for the removal of toxic dyes from wastewaters
Manuscript ID: RSOS-200438

The editor assigned to your manuscript has now received comments from reviewers. We would like you to revise your paper in accordance with the referee and Subject Editor suggestions which can be found below (not including confidential reports to the Editor). Please note this decision does not guarantee eventual acceptance.

Please submit your revised paper before 16-May-2020. Please note that the revision deadline will expire at 00.00am on this date. If we do not hear from you within this time then it will be assumed that the paper has been withdrawn. In exceptional circumstances, extensions may be possible if agreed with the Editorial Office in advance. We do not allow multiple rounds of revision so we urge you to make every effort to fully address all of the comments at this stage. If deemed necessary by the Editors, your manuscript will be sent back to one or more of the original reviewers for assessment. If the original reviewers are not available we may invite new reviewers.

Royal Society of Chemistry
Thomas Graham House

Science Park, Milton Road
Cambridge, CB4 0WF
Royal Society Open Science - Chemistry Editorial Office

RSC Associate Editor:
Comments to the Author:
(There are no comments.)

RSC Subject Editor:
Comments to the Author:
(There are no comments.)

Reviewers' Comments to Author:
Reviewer: 1

Comments to the Author(s)

The article is technically good. I suggest to put more attention to Fig. 7 and discuss in detail the mechanism. In addition to obvious "electrostatic interaction, hydrogen bonding and pi-pi interactions", other interactions could be between O- and N-doped atoms, which frequently present in carbon matrix. The role of silica in adsorption process should be emphasized. Main recent books in dye removal by nanocomposites, hybrids and nanocomposites in general should be added and cited.

Reviewer: 2

Comments to the Author(s)

This is a very interesting paper about a very nice approach to composite materials which would be expected to have very positive properties. It uses a large volume waste material from sugar production as a carbon source, which is an intelligent way to produce the materials, improving the literature routes to other similar materials.

There are a few fairly minor matters I would like the authors to address before publication:

The CSC600 material seems unusual – I would expect the C content to decrease as T increases.

Thermal analysis traces of mass loss suggest that it should be slightly lower.

Thermal treatment (TGIR data) – clearly also some organics evolved at low (200-250oC) temperatures. Do the authors have any insights into what these may be? Presumably

furaldehydes and similar dehydration products, as the traces seem very similar to molasses itself.

Abstract – mesopore volumes greater than 90% (presumably of the total pore volume?)

Experimental – how much water was added to the molasses? What is the water content of the molasses?

There must be a significant amount of supernatant liquid after mixing – is this removed before oven treatment?

XRD graphite at 52 and 43 (not 25 and 43)?

Reviewer: 3

Comments to the Author(s)

This manuscript is needed a major revision and further review. The comments are given below.

Limitation of the work, novelty and contributions should be highlighted more.

Following references should be included in the introduction part for more readable, about various carbon nanostructures and its structure-properties. *Int. J. Hydrogen Energy* 45 (2020) 7716-7740; *Carbon* 149 (2019) 693-721; *Synthetic Metals* 159 (2009) 1934-1939; *Microchemical Journal* 149 (2019) Article 103985; *Small methods* 2 (2018) Article 1800050; *Scripta Materialia* 58 (2008) 1010-1013; *J. Environ. Manage.* 238 (2019) 25-40; *ChemistrySelect* 4 (2019) 5322-5337; *Int. J. Hydrogen Energy* 44 (2019) 10453-10472; *J. Environ. Manage.* 254 (2020) Article 109747; *Chemical Eng. Journal* 309 (2017), 151-158; *J. Environ. Manage.* 250 (2019) Article 109486; *Chemistry - An Asian J.* 13 (2018) 3561-3574.

Please compare and discuss the differences between diffractions in Figure 3B.

The growth mechanism for the formation of CSCs should be explained in more detail based on the chemistry concept.

characterization data is needed for this work. It includes HRTEM images, SAED, EDS, XPS and Raman data.

DTG curves should be included in TGA curves to see accurate solvent removal and thermal degradation temperatures. Please tabulate the different thermogravimetric parameters (Tonset, Td, Tg, ΔT and % char).

Theoretical maximum adsorption capacity should also be calculated and compare with experimental maximum adsorption capacity.

What about the cycle tests of adsorption process for congo red removal? What is the effect of pH on adsorption performance?

The authors should update the following references in the results part, about the different techniques used for removal of chemical pollutants: *Chem Eng. J* 385 (2020) Article 123787; *Small methods* 2 (2018) Article 1800050; *Separation and Purification Tech.* 210 (2019) 850-866; *J. Environ. Manage.* 265 (2020) 110504; *J. Nanosci. and Nanotech.* 10 (2010) 7951-7957; *Chemosphere* 239 (2020) 124766.

What would be the effect of different conc. of molasses and higher carbonization temperature (>800 degC) on the adsorbent performance.

In the conclusion part, the authors should mention about the future direction of the work.

Author's Response to Decision Letter for (RSOS-200438.R0)

See Appendix A.

RSOS-200438.R1 (Revision)

Review form: Reviewer 1

Is the manuscript scientifically sound in its present form?

Yes

Are the interpretations and conclusions justified by the results?

Yes

Is the language acceptable?

Yes

Do you have any ethical concerns with this paper?

No

Have you any concerns about statistical analyses in this paper?

No

Recommendation?

Accept as is

Comments to the Author(s)

After these corrections, the manuscript can be now published as it is.

Decision letter (RSOS-200438.R1)

Dear Dr Supanchaiyamat:

Title: Graphitic mesoporous carbon-silica composites from low value sugarcane by-products for the removal of toxic dyes from wastewaters

Manuscript ID: RSOS-200438.R1

It is a pleasure to accept your manuscript in its current form for publication in Royal Society Open Science. The chemistry content of Royal Society Open Science is published in collaboration with the Royal Society of Chemistry.

RSC Associate Editor:
Comments to the Author:
(There are no comments.)

RSC Subject Editor:
Comments to the Author:
(There are no comments.)

Reviewer(s)' Comments to Author:
Reviewer: 1

Comments to the Author(s)
After these corrections, the manuscript can be now published as it is.

Appendix A

Department of Chemistry
Faculty of Science
Khon Kaen University
Khon Kaen 40002
Thailand
Email: nontsu@kku.ac.th

August 29, 2020

Dear Editor in Chief,

Please find attached our revised article entitled “Graphitic mesoporous carbon-silica composites from low value sugarcane by-products for the removal of toxic dyes from wastewaters”, for publication in Royal Society Open Science. We believe that we have addressed the referees’ comments and have attached the necessary changes below. The authors would like to thank the referees for their comments and suggestions. We feel these changes have considerably improved the quality of the manuscript. As requested a corrected version of the paper has been uploaded with this letter. Please find response to referees comments below.

Response to Referees

Reviewer: 1

Comments to the Author(s)

The article is technically good. I suggest to put more attention to Fig. 7 and discuss in detail the mechanism. In addition to obvious "electrostatic interaction, hydrogen bonding and pi-pi interactions", other interactions could be between O- and N-doped atoms, which frequently present in carbon matrix.

The authors agree with the referee and the interactions of O atoms have been further highlighted in the text. “Importantly, the unique range of functional groups present in carbon matrix of molasses-based CSC such as C-O-C, C-OH, C=O and COO- could also play a vital role in the adsorption mechanism of the dye. The proportions of these

functionalities clearly demonstrate significant differences over other CSC materials in the literature. This is especially true for CSC600 and CSC800, where XPS demonstrates the formation of C=O at 600 °C, while C-O-C functional groups are dominate at 800 °C.”

The role of silica in adsorption process should be emphasized.

The role of silica has now been discussed within the manuscript. “In the case of methylene blue, it was noted that silica might play a role in the cationic dye adsorption as it was reported that mesoporous silica could selectively adsorb cationic molecules. This is due to the electrochemical interaction between the cationic molecules and the negative charges on the silica surfaces. However, the effect of silica adsorption in the composites should be minimal as the SEM-EDS images showed uniform coverage of carbon onto silica. Although silica may not be directly involved in the adsorption process, the mesoporous structure it provides is likely to enable the rapid diffusion of dye molecules into the CSC framework.”

Main recent books in dye removal by nanocomposites, hybrids and nanocomposites in general should be added and cited.

The authors would like to thank the reviewer for constructive comments. Additional information from recent books in dye removal have been included in the manuscript as suggested.

Reviewer: 2

Comments to the Author(s)

This is a very interesting paper about a very nice approach to composite materials which would be expected to have very positive properties. It uses a large volume waste material from sugar production as a carbon source, which is an intelligent way to produce the materials, improving the literature routes to other similar materials.

The authors would like to thank the reviewer for their time for reviewing this manuscript.

There are a few fairly minor matters I would like the authors to address before publication:

The CSC600 material seems unusual – I would expect the C content to decrease as T increases. Thermal analysis traces of mass loss suggest that it should be slightly lower. Thermal treatment (TGIR data) – clearly also some organics evolved at low (200-250oC) temperatures. Do the authors have any insights into what these may be? Presumably furaldehydes and similar dehydration products, as the traces seem very similar to molasses itself.

The TGIR data showed that there were two main decompositions of molasses at 200-250 °C and around 700 °C, which released gases such as water, CO₂ and other dehydration products (including products such as 5-hydroxymethylfurfural). The CSC800 showed the lowest C content as both decompositions would have had occurred. However, the fact that CSC600 has higher carbon content than CSC400 does seem unusual. CHN analysis showed that CSC400 comprised of higher amounts of hydrogen and other elements compared to CSC600. Also the DTG of CSC600 (ESI-Figure S2B) showed that the carbon layer of the CSC600 decomposed in two steps while the others exhibited one-step decomposition. These facts might contribute to the higher C content of CSC600, however more work needs to be done to elucidate this matter.

Abstract – mesopore volumes greater than 90% (presumably of the total pore volume?)
Indeed, the percentage of the mesoporosity was calculated from the mesopore volume divided by the total pore volume, as such the phrase “(calculated from the total pore volume)” was added to the abstract.

Experimental – how much water was added to the molasses? What is the water content of the molasses?

The molasses was used as received from the sugar plant and the water content of the molasses according to the TGA result was approximately 18% and this has been added to the experimental part of the manuscript.

There must be a significant amount of supernatant liquid after mixing – is this removed before oven treatment?

Yes, indeed. The supernatant liquid was removed by decantation followed by a filtration prior to the drying in the oven.

XRD graphite at 52 and 43 (not 25 and 43)?

The two diffraction peaks of graphitic carbon observed at 25° and 43° which correspond to (002) and (100) or (101) planes of crystalline hexagonal graphite lattice (according to references 45-48). A peak at 55° corresponding to (004) plane of graphite lattice which can be clearly seen in high purity graphite (references 45 and 48) is not observed in our composites.

Reviewer: 3

Comments to the Author(s)

This manuscript is needed a major revision and further review. The comments are given below.

Limitation of the work, novelty and contributions should be highlighted more.

The novelty of the work has been further expressed throughout the text including demonstrating that the CSC materials have distinct changes in functionality as a function of preparation temperature. XPS demonstrated that the functional nature of the carbon layer on the surface of the materials differed to other CSC materials in the literature. As the referee recommended for the authors to undertake XPS we would like to thank them for this suggestion.

Following references should be included in the introduction part for more readable, about various carbon nanostructures and its structure-properties. Int. J. Hydrogen Energy 45 (2020) 7716-7740; Carbon 149 (2019) 693-721; Synthetic Metals 159 (2009) 1934-1939; Microchemical Journal 149 (2019) Article 103985; Small methods 2 (2018) Article 1800050; Scripta Materialia 58 (2008) 1010-1013; J. Environ. Manage. 238 (2019) 25-40; ChemistrySelect 4 (2019) 5322-5337; Int. J. Hydrogen Energy 44 (2019) 10453-10472; J. Environ. Manage. 254 (2020) Article 109747; Chemical Eng. Journal 309 (2017), 151-158; J. Environ. Manage. 250 (2019) Article 109486; Chemistry - An Asian J. 13 (2018) 3561-3574.

A paragraph regarding the various carbon nanostructures and its properties has been added to the introduction of the manuscript with a citation of relevant recommended references.

“Carbon-based materials have found use in a wide range of application owing to their unique 3D structure, high surface area, thermal and chemical stability and decent conductivity. Their unique characteristics have enable a number of carbonaceous materials including carbon black, graphene, graphene oxide, carbon fibre and carbon nanotubes to be utilised in composites/hybrid materials in order to enhance their properties in various applications such as catalysis, nanoelectronics and sensors. ”

Please compare and discuss the differences between diffractions in Figure 3B.

The discussion of the differences of the XRD patterns of CSCs has been revised as follows.

“Figure 3B shows the XRD patterns of the CSCs. A peak at 22° which is characteristic of the cristobalite structure of silica was observed for all spectra. The intensity of this broad peak was less intense in CSC600, confirming the higher carbon to silica ratio as seen in TGA and CHN analysis. Interestingly, two diffraction peaks of graphitic carbon at 25° and 43° which correspond to (002) and (100) or (101) planes of crystalline hexagonal graphite lattice respectively, were clearly observed. This confirms the FT-IR results that the carbonisation of this composite at 800 °C gives rise to the graphitic carbon structure, which is in accordance with previous reports of CSC from sucrose.”

The growth mechanism for the formation of CSCs should be explained in more detail based on the chemistry concept.

The formation of the carbon coated onto silica has been explained in more details as stated in the following text.

“Synthesis of CSCs was simply performed through the coating of molasses onto the silica using Liquid antisolvent precipitation (LAP) process. LAP is based on the change of supersaturation caused by mixing the solution and the antisolvent . The process requires two miscible solvents and the chemical must dissolve in the solvent but not in the antisolvent. In the case of molasses based CSC, water in molasses is the solvent, while ethanol acts as an antisolvent. The addition of ethanol to the water-molasses suspension

thus causes the precipitation of molasses onto silica particles, resulting in a desirable coating effect.”

characterization data is needed for this work. It includes HRTEM images, SAED, EDS, XPS and Raman data.

The EDS results of the materials have already been included in the first version of the manuscript (page 20). The XPS and Raman data have now been added (page 15-17).

DTG curves should be included in TGA curves to see accurate solvent removal and thermal degradation temperatures. Please tabulate the different thermogravimetric parameters (Tonset, Td, Tg, ΔT and % char).

DTG curves have been included in Figure S2 of the ESI and the TGA parameters have been presented in Table S3.

Theoretical maximum adsorption capacity should also be calculated and compare with experimental maximum adsorption capacity.

The theoretical adsorption capacities have already been included in Table 4 (for congo red) and Table 5 (for methylene blue) in the first version of the manuscript. The comparison of theoretical and experimental adsorption capacities of both congo red and methylene blue has now been added in the discussion as follow.

“It was noted that the experimental adsorption capacity of CSC600 was at 291 mg/g (Table 6), which was close to the theoretical monolayer adsorption capacity (290 mg/g). The highest monolayer adsorption capacity was obtained from CSC800 at 390 mg/g, while the experimental adsorption capacity for this material was $Q_{exp}=455\text{mg/g}$, which was attributed to the highly aromatic carbon surface of the adsorbent. In contrast, CSC400 demonstrates an experimental adsorption of $Q_{exp}= 381 \text{ mg/g}$, which far exceeded the predicted 269 mg/g monolayer adsorption capacity. This thus demonstrates that CSC400 is likely to undertake multilayer adsorption.”

“The experimental adsorption capacities of both CSC600 and CSC800 were very close to the theoretical monolayer adsorption capacities, whilst CSC400 showed much higher experimental value, indicating that the adsorption in the latter should consist of both monolayer and multilayer adsorption.”

What about the cycle tests of adsorption process for congo red removal? What is the effect of pH on adsorption performance?

The effect of pH on the adsorption performance of both methylene blue and congo red (pH 2, 5, 7 and 10) has now been investigated and a discussion of the results has been added as follows.

“The effect of pH on the efficiency of the CSC adsorbents on the adsorption of congo red was also studied at the pH values of 2-10. It was observed that the removal of congo red is more efficient in acidic solution in which congo red dissociates to sulfonic group ($-\text{SO}_3^-$) (ESI, Figure. S13). The carbonaceous surface of CSC is positive in acidic medium, hence the induced electrostatic interactions between the dye and the adsorbent favoured the adsorption of the dye onto the CSCs. Similar results have been obtained by for the adsorption of congo red onto activated carbons. CSC materials exhibited high percentage removal of congo red in acidic solution at 97-98% and similar value was also observed for graphene quantum dots whose preparation would require more complex method. It was noted that the CSCs prepared at different temperatures showed no significant differences in the influence of pH on dye removal.”

The reusability of the materials including cycle tests, the burn-off of carbon coating at the end of use and the reuse of silica to produce the composites will be investigated in our future work. This has been highlighted within the manuscript.

The authors should update the following references in the results part, about the different techniques used for removal of chemical pollutants: Chem Eng. J 385 (2020) Article 123787; Small methods 2 (2018) Article 1800050; Separation and Purification Tech. 210 (2019) 850-866; J. Environ. Manage. 265 (2020) 110504; J. Nanosci. and Nanotech. 10 (2010) 7951-7957; Chemosphere 239 (2020) 124766.

The relevant references have been included in the results parts to highlight various techniques in chemical pollutants removal.

What would be the effect of different conc. of molasses and higher carbonization temperature (>800 degC) on the adsorbent performance.

The study of the effect of ratio of silica to molasses has actually been investigated prior to the investigation of the effect of pyrolysis temperature of CSCs. The molasses to silica ratio of 4/1 by weight was determined as the optimum ratio that had the highest incorporated molasses onto the silica. This has now been explained in the main text and the details of this study were added to the ESI.

The higher carbonization temperature is likely to widen the pores on the carbon layer as seen in several occasions in carbon/activated carbon. This would affect the adsorption performance but it is highly dependent on the adsorbates. Future work is needed to elucidate this matter.

In the conclusion part, the authors should mention about the future direction of the work. **Future work which include the reusability of these CSC materials including cycle tests, the burn-off of the carbon coating at the end of life to generate energy and the reuse of the silica support and has been added to the conclusion as suggested.**

Many thanks for taking the time to review our changes and we hope that our manuscript will be accepted in the near future. All correspondence concerning this paper should be addressed to Dr. Nontipa Supanchaiyamat (e-mail: nontsu@kku.ac.th).

Yours sincerely,

Nontipa Supanchaiyamat